Fluctuations of continuous soil moisture evaporation under different rainfall conditions during the growing period of the non-monsoon season, the eastern Loess Plateau

http://orcid.org/0000-0002-9767-9667 Sun Congjian 1 2 suncongjian@sina.com
Meng Sitong 1 2
Chen Wei 1 2
1 School of Geographical Sciences, Shanxi Normal University , Taiyuan , China
2 Research Center for Ecological Environment in the Middle Reaches of the Yellow River, Shanxi Normal University , Taiyuan , China
Oehlmann Jörg
Electronic publication date: 2024 Nov 22
Publication date: 2024
Volume: 12
Electronic Location ID: e18514
Received 2024 Jul 9; Accepted 2024 Oct 21
Copyright: © 2024 Sun et al.
Copyright year: 2024
Copyright holder: Sun et al.
License: This is an open access article distributed under the terms of the Creative Commons Attribution License, which permits unrestricted use, distribution, reproduction and adaptation in any medium and for any purpose provided that it is properly attributed. For attribution, the original author(s), title, publication source (PeerJ) and either DOI or URL of the article must be cited.
License URL: https://creativecommons.org/licenses/by/4.0/

Keywords: Hydrogen and oxygen isotopes, Seasonal and spatial distribution, Evaporation loss, Controlling factor

Funding: Fundamental Research Program of Shanxi Province 202303021221154 This work was supported by the Fundamental Research Program of Shanxi Province (Grant No. 202303021221154). The funders had no role in study design, data collection and analysis, decision to publish, or preparation of the manuscript.

==============================
Soil moisture is an important link between material and energy exchange between the land and atmosphere, and its evaporation loss is crucial to sustainable development of agriculture. Based on observations of long-term stable isotopes of soil moisture in the eastern Loess Plateau (ECLP) during the non-rainy season growing period, this study systematically explored soil water evaporation loss at different soil depths using the Craig–Gordon model and revealed the internal relationship between soil evaporation loss and environmental elements. Main findings included: (1) The soil moisture content showed a gradual decreasing trend, with a weak soil moisture δ18O fluctuation appearing in April, whereas a stronger fluctuation was observed in June. (2) A significant vertical spatial heterogeneity was observed in soil moisture δ18O of each soil layer. Enriched soil moisture δ18O values appeared in the 0–20 cm soil layer, and the minimum value appeared in the 40–60 cm soil layer. (3) A significant spatial and temporal heterogeneity was observed in the soil moisture evaporation loss fraction (f) (0–23.35%), with weaker values at the beginning of the study period and larger values between mid-late May and mid-June. The largest soil evaporation loss was observed in the 0–20 cm soil layer (average value of 8.97%), a fluctuating decreasing trend appeared with increasing soil depth. (4) Regional soil moisture evaporation loss was positively correlated with regional air temperature (T) and potential evapotranspiration (ET0) and negatively correlated with soil water content (SWC) and relative humidity (RH). The correlation between soil moisture evaporation loss and environmental elements gradually weakened with increasing soil depth. (5) The environmentally driven model of continuous evaporation of soil moisture was suitable for larger amounts, especially for the surface soil layers. The results of this study have important implications for water resource management, ecosystem stability, and sustainable regional agriculture in the ECLP.

Introduction

As an important limiting factor for vegetation growth and ecosystem stability, soil moisture is a key component of regional water resources as well as an important link to the exchange of matter and energy between the atmosphere and the Earth’s surface (Jun et al., 2021). Dynamic variation in soil moisture strongly influences regional soil erosion, solute transport, land-atmosphere interaction, and groundwater recharge and discharge patterns and is also a driving force for regional soil formation processes and a series of landform changes (Scanlon et al., 2006; Chen et al., 2022b; Tan et al., 2016; Goldin, 2016). Soil moisture is highly sensitive to variations in regional environmental elements such as precipitation, evaporation, topography, land use, vegetation, and soil properties (Yu et al., 2018; Jungandreas, Hohenegger & Claussen, 2023). As an important form of soil moisture consumption (Qiu-Wen et al., 2022), soil moisture evaporation loss directly determines the efficiency of converting natural precipitation into effective soil moisture and plays an important role in the vertical movement of regional soil moisture and the spatiotemporal distribution of water resources (Tugwell-Wootton et al., 2020; Tesfuhuney et al., 2022), especially in arid and semi-arid areas with relatively scarce water resources. The systematic study of the spatial and temporal dynamic changes in soil water and the quantitative assessment of soil moisture evaporation rate have attracted increasing attention, which also has important theoretical and practical significance for the rational optimal allocation of regional water resources, prevention of soil erosion, and stability of agricultural ecosystems (Maren & Christiane, 2019; Ala-aho et al., 2018).

Recently zero-flux plane, numerical simulation, water balance, and laser methods, and the stable isotope tracing technology, have been widely used in studies on quantitative soil water, resulting in a series of important guiding results (Yang et al., 2023; Chakraborty et al., 2022; Baalousha et al., 2022). Based on the zero-flux surface method, Yang et al. (2023) explored the regional water balance of the aeration zone in arid irrigation areas and quantitatively estimated the amount of soil moisture recharged to shallow groundwater. In the western United States, researchers simulated dynamic changes in soil moisture under different crops based on the HYDRUS-1D model (Chakraborty et al., 2022). Combined with field observations and numerical simulations, Baalousha et al. (2022) systematically explored the dynamic changes of evapotranspiration and soil moisture content in arid areas. However, most of these studies were conducted at experimental sample sites and were strongly dependent on observation instruments (Tesfuhuney et al., 2022; Maren & Christiane, 2019; Ala-aho et al., 2018). In addition, relatively high costs and errors make it difficult to generalize these methods widely, especially in the systematic exploration of the process and mechanism of soil moisture evaporation in typical areas under complex terrain conditions (Skrzypek et al., 2015). Stable hydrogen and oxygen isotopes are widely used as tracers in natural water bodies, which can more accurately and quickly reveal information on specific soil hydrological processes, such as infiltration, evaporation, transpiration, mixing process, and soil water migration processes (Zimmermann et al., 1966; Rothfuss et al., 2012; Fekete et al., 2016; Zhu et al., 2019). The Craig–Gordon (C–G) model was widely used and effective method to quantify water evaporation loss (Xiao, Wei & Wen, 2018). For example, in the Pampas Plains of Argentina (Quiroz Londoño et al., 2020), studies were conducted on the quantitative estimation of evaporation loss in shallow lakes of the temperate zone based on the C-G model, and the monthly mean range of evaporation loss on the lake surface was approximately 20% to 25%. In Canada (Gibson et al., 2017), the quantitative estimation of the evaporation loss of six lakes based on the C-G model indicated that approximately 18% of the lake water evaporated each year and showed a decreasing trend from south to north. In China, a quantitative estimation of the evaporation loss of water samples along the line of the South–North Water Diversion project showed significant seasonal heterogeneity between July and April (Chen & Tian, 2021). Since then, researchers have made continuous corrections to the C-G model and have made it an important research method to explore the process of regional evapotranspiration (Lu et al., 2023), which has realized the quantitative estimation of soil water evaporation in a certain period of time under certain assumptions (Skrzypek et al., 2015). Based on the improved C-G model, quantitative studies have been conducted on regional soil evaporation characteristics in typical areas, and effective results were obtained with certain practical application values (Dubbert et al., 2013; Grevengoed et al., 2023; Al-Oqaili et al., 2020; Chen et al., 2022a; Wei et al., 2018; Piayda et al., 2017; Qiu et al., 2023; Lyu & Wang, 2021; Swarowsky et al., 2011). According to systematic analysis of soil moisture evaporation in the northwestern United States, approximately 6–14% of the soil moisture in this region is caused by regional evaporation (Grevengoed et al., 2023). In the mid-northwestern United States (Al-Oqaili et al., 2020), the estimation results of soil water loss in regional potato fields based on the Craig–Gordon model showed that the evaporation loss at the inter-row position had a relatively smaller value, with a maximum evaporation loss of only 18%. Long-term observations of urban woodland in the North China Plain showed that the ratio of vegetation transpiration and land evaporation in this area fluctuated between 0.21 and 0.95 (Chen et al., 2022a). Currently, an increasing number of studies are focusing on the factors controlling the regional evaporation process, and related studies are being conducted in certain regions (Piayda et al., 2017; Qiu et al., 2023; Lyu & Wang, 2021). In North America, an observational study of the soil evaporation response after precipitation events showed that soil moisture variability and evaporation losses in surface soil (particularly at 10 cm depth) showed a notable response to precipitation (Piayda et al., 2017). In a study of soil moisture dissipation in plantations in arid oasis areas, high temperatures and sufficient precipitation resulted in substantial evaporation of soil water in the summer (Qiu et al., 2023). In subtropical plantation areas, dynamic changes in surface soil moisture evaporation loss are closely related to increases in wind speed (Lyu & Wang, 2021). In the Mediterranean region, monitoring of soil evaporation loss under cork oak forests showed that woodland vegetation cover can effectively reduce evaporation losses, especially in deep soil (Swarowsky et al., 2011). Soil evaporation loss presents significant spatial and temporal heterogeneity and is regulated by regional environmental factors (air temperature, precipitation, wind speed, atmospheric humidity, and surface cover) (Dubbert et al., 2013; Grevengoed et al., 2023; Al-Oqaili et al., 2020; Chen et al., 2022a; Wei et al., 2018; Piayda et al., 2017; Qiu et al., 2023; Lyu & Wang, 2021; Swarowsky et al., 2011). However, existing studies have mainly been conducted under laboratory and semi-experimental conditions and have not systematically explored continuous evaporation under natural conditions, especially for arid and semi-arid areas with complex terrain, arid climate, large evaporation, and a relative shortage of water resources (Li et al., 2023). Quantitative research on continuous soil evaporation under natural conditions in arid and semi-arid areas needed to be further strengthened.

The eastern Loess Plateau (ECLP) is located in a climate zone between the semi-arid and semi-humid regions, with a complex climate, changeable terrain, and a lack of surface water resources. Soil moisture is the main limiting factor in regional agricultural production and ecological construction, particularly during the non-monsoon season (Wang et al., 2021; Zhang et al., 2023). During the non-monsoon season, rapid warming and scarce precipitation exacerbate regional drought, resulting in stronger soil evaporation losses. Fast-growing regional vegetation showed a strong dependence on soil moisture during this period. Soil moisture evaporation and loss would restrict the normal growth of plants especially in summer and early summer, further leading to crop dysplasia and even reduced production, which considerably affects the development of agriculture and food security (Kim et al., 2022; Wang et al., 2023; Wu et al., 2022; Li et al., 2023). Since 2000, the large-scale activities of returning farmland to forest and grassland in the ECLP has led to the increasing complexity of regional soil water migration, especially in the non-monsoon period of drought and less rain. It is not clear about the spatial and temporal heterogeneity of soil moisture and its evaporation loss in typical agricultural land under the background of farmland conversion (Gou & Zhu, 2021; Li et al., 2018). There is an urgent need to explore regional soil water fluctuation characteristics and estimate soil water evaporation losses of the non monsoon season for regional ecological and environmental protection and sustainable development.

The hilly and gully areas of the ECLP were selected as typical study areas. Based on long-term continuous observation and measurement of stable isotopes in soil water during the non-rainy season, this study analyzed the spatial and temporal variation of stable isotopes in soil moisture during the non-rainy season, quantitatively evaluated soil evaporation loss at different soil depths, and explored the internal relationship between soil evaporation loss and major regional environmental elements during the study period. The results of this study have important implications for water resource allocation and sustainable agricultural development in the growing season during the non-monsoon season on the eastern Loess Plateau.

Materials and methods

Study area

The study area was located in an open farmland with the implementation of the policy of returning farmland (112°66′E, 37°74′N) in the ECLP, 772 m above sea level (Fig. 1). Mountains surrounded the experimental area on three sides, with terrain high in the north and low in the south (Li et al., 2022b). The climate pattern of the study area belongs to the continental temperate monsoon climate, and the annual average temperature is 9.5 °C, with large interannual and seasonal temperature fluctuations (the extreme maximum temperature is 37 °C, and the minimum temperature is −22.8 °C). Precipitation in the study area was relatively rare and concentrated, with an average annual precipitation of 468.4 mm, and was mostly concentrated in summer. The average annual potential evaporation in the study area was 1,774.5 mm (Sun et al., 2020a). Under the strong influence of the East Asian monsoon, the rainy season begins in the study area at the end of June. During the rapid growth period of plants before the rainy season, there is scarce precipitation, strong soil water evaporation, and large plant water demand. This is the most prominent contradiction between water resource supply and demand in this region, especially for agricultural systems (Xiang et al., 2020). Therefore, the non-monsoon growth period from April 1 to July 1 was selected as the study period. The average temperature in this period was 12 °C, the average precipitation was only 57.5 mm (13–15% of the year (Sun et al., 2020b)), and the potential evaporation was 517.4 mm (29% of the year). The soil type in the study area is mainly brown soil with loose soil quality and a deep soil layer (Guo et al., 2019). In addition, the study area is a typical dryland agricultural area, and crop planting has a large demand for water resources. The main crops include winter wheat, spring wheat, and spring corn (Liu et al., 2022).

Figure 1 Location and geographical overview of the study site.

(A) The location of the study area (image source: The Resource and Environmental Science Data Platform, https://www.resdc.cn). (B) The surface soil moisture content of the study area in 2022. (C) The average temperature, monthly average temperature, and monthly precipitation of the study area in the past 5 years. (D) High altitude photos of the test site location taken by drones.

Information on sampling and measurement

Soil water samples were collected from the test site of the Ecological Environment Research Center of the Middle Reaches of the Yellow River at Shanxi Normal University. In this study, soil samples were collected daily continuously in the study area from April 1, 2023, to July 1, 2023, and sampling activities were conducted daily from 18:00 to 19:00. Three parallel quadrats, 1 × 1 m, were selected in each sampling activity for further study. Soil samples were obtained using a 38-mm diameter handheld soil drill. To avoid the influence of air on the soil samples, the 1–2 cm soil surface was removed using a shovel before each sampling. Soil samples were collected from a depth of 0–100 cm at 20 cm intervals. The different samples from the same layer of three parallel quadrats were mixed to create a composite sample. After sampling, part of the soil was loaded into a 10 mL glass bottle, quickly sealed with a Parafilm membrane, and brought back to the laboratory for soil water extraction. The other part was layered and packed into a sealed bag to measure soil moisture content (SWC). Atmospheric precipitation samples were collected during precipitation events. A rain bucket was placed near the sample point to collect the precipitation. Precipitation samples were sealed immediately after precipitation (stored in a 10 mL sampling tube and sealed with parafilm), and the corresponding meteorological elements (precipitation time, real-time precipitation, temperature, relative humidity, and wind speed) were simultaneously recorded. During the sampling period (April 1, 2023–July 1, 2023), 460 soil samples and 16 precipitation samples were collected.

During the sampling period, meteorological data of the study area (air temperature (T), precipitation amount (P), relatively humidity (RH), wind speed (WS)) were downloaded from the https://cds.climate.copernicus.eu/datasets/reanalysis-era5-land?tab=download (Taiyuan airport), and the ET0 data were calculated by using the Penman-Monteith formula calculator.

The stable isotopes of the water samples were measured using a Liquid Water Isotope Analyzer (Los Gatos Research, Inc., San Jose, CA, USA, Type: GLA431-LWIA) at the Ecological Environment Research Center in the middle reaches of the Yellow River at Shanxi Normal University. The measurement precision was 0.2‰ and 0.03‰ for δ2H and δ18O, respectively. The Vienna Standard Mean Ocean Water (V-SMOW2) was used to revise the measured results (δ-values, shown as Eq. (1)). To reduce cross-contamination, the number of samples measured in each batch was controlled at approximately 35. Before each batch of samples was measured, a standard sample was prepared, the syringe was cleaned, and the diaphragm was replaced. Each sample was retested six times, and the LWIA Post Analysis software (v. 3.1.0.9) was used to reanalyze the measured data. The average of the last four needles of the measured data was selected as the output value (Sun et al., 2020c).

(1) δ(‰)=(Rsample−RstandardRstandard)×1000%

where Rsample is the ratio of 2H or 18O in the measured sample, and Rstandard is the ratio of 2H or 18O to the Vienna standard mean ocean water.

Lc-excess

In this study, the intensity of soil water evaporation relative to local precipitation was characterized by calculating the Lc-excess for each soil water sample. The linear relationship of δ2H and δ18O is defined as the local atmospheric waterline (LMWL) and soil waterline (SWL) in precipitation and soil water, respectively. In contrast, the deviation between LMWL and SWL can be defined by Lc-excess, which can be used to determine the evaporative fractionation process of different water bodies (Hasselquist et al., 2018; Hervé-Fernández et al., 2016).

(2) Lc−excess=δ2H−aδ18O−b

where a and b represent the slope and intercept of LMWL, respectively, and δ2H and δ18O values represent the hydrogen and oxygen isotope values in the sample, respectively. Generally, the annual average value of lc-excess in precipitation is 0‰, whereas the water body lc-excess is less than 0‰ (and is below LMWL), indicating that the water body is affected by strong evaporation. However, lc-excess showed a positive value in the water samples, indicating that the water samples may be affected by water sources other than precipitation (Landwehr, Coplen & Stewart, 2014).

Quantifying soil moisture evaporation using the C-G model

Soil evaporation loss f under continuous evaporation conditions from April 1, 2023, to July 1, 2023, was quantified using the nonstationary model in the Craig–Gordon model (non-steady-state model) (assuming that the stable isotope composition of soil water is altered by isotopic fractionation during evaporation) (Skrzypek et al., 2015; Li et al., 2023; Craig, Gordon & Horibe, 1963). The equation is as follows:

(3) f=1−[δL(soil)−δ∗δP(soil)−δ∗]1m

where δP(soil) is the initial value of the soil-water sample oxygen isotope composition (δ18OP(soil)), δL(soil) is the final value of the soil-water sample oxygen isotope composition, and δ* is the limiting factor for oxygen isotope enrichment (Gat, 1978). The calculation method is as follows:

(4) δ∗=h×δA+εh−ε1000

where h is the average relative humidity (fraction), ε is the total isotope fractionation (‰), and δA is the stable isotopic composition of water in ambient air (‰). δA is generally determined based on the isotopic composition of precipitation samples (Gibson & Reid, 2014). The equation is as follows:

(5) δA=δrain−ε+α+

where δrain is the measured value of the precipitation isotope at the beginning of each continuous evaporation, and ε+ and α+ are the equilibrium enrichment factor and equilibrium fractionation factor that change with temperature, respectively. This can be calculated using the following equation:

(6) ε+=(α+−1)×1000

(7) α(18O)+=e[−7.685∗10−3+6.7123∗T−1−1.6664∗103∗T−2+0.35041∗106∗T−3]

where temperature (T) is expressed in Kelvin.

(8) ε=ε+α++εk

where ε is the total isotopic fractionation factor (‰) and εk is the kinetic isotopic fractionation factor (‰).

(9) εk=(1−h)×nθCD

where n is a constant, which is related to the correlation between the molecular diffusion resistance and molecular diffusion coefficient and is usually considered to be 1 for non-mobile air layers (e.g., soil moisture evaporation and plant transpiration), θ is the ratio of the molecular diffusion coefficient to the total diffusion fractionation coefficient and is generally considered to be 1 for soil moisture evaporation. CD is a parameter that describes the diffusion efficiency of the molecule and has a value of 25.1‰ and 28.5‰ for hydrogen and oxygen, respectively (Crusius & Thomson, 2000).

(10) m=h−ε10001−h+εk1000

where m represents the correlation between the isotopic compositions of the local evaporated water vapor and liquid water, also known as the enrichment slope of the local evaporated water line (Wendong et al., 2023).

The redundancy analysis and the multiple linear regression analysis

The redundancy analysis (RDA) was used to explore the relationship between soil moisture evaporation loss and the main influencing factors using RStudio4.3.2 software (Capblancq et al., 2018; Du et al., 2021). In this study, the response matrix was the evaporation loss matrix at different soil depths and the explanation matrix was a matrix composed of environmental factors. Based on Monte Carlo random permutation tests, each ranking axis was tested individually.

Multiple linear regression was used to further fit the relationship between the continuous soil moisture evaporation loss and environmental factors (Sixtine et al., 2021; Nejatian et al., 2023). In this study, the multiple linear regression functions of the explained variable y (soil moisture evaporation loss) and the variables X2, X3, …, Xk (the main influencing factors) were as follows:

(11) y=α+β1x1+β2x2+β3x3+...+βkxk

where βj (j = 1, 2,…, k) is the parameter of the model; k is the number of explanatory variables. Below, R2 was used to test the goodness of fit, and a t-test was used to test the significance of the regression parameters.

Results and discussion

Information for regional air temperature, humidity, potential evaporation, and precipitation

During the study period, the daily air temperature (T) ranged from 4.7 °C to 35 °C, with an average value 22 °C. The lowest and highest air temperatures occurred on April 11 and June 22, respectively. Air temperature showed an overall upward trend, with a rapidly increasing trend appearing between May and June. During the study period, the daily average relative humidity (RH) varied between 11% and 86%, with a mean value of 34%. Relatively lower RH values were observed in June, and relatively higher values were observed in May (Fig. 2). The daily potential evapotranspiration (ET0) during the study period ranged from 2.2 to 21.4 mm, with a mean value of 5.95 mm, and the total potential evapotranspiration was 549 mm. A relatively stronger ET0 was observed in June.

Figure 2 Daily variations of air temperature (T), relative humidity (RH), potential evaporation (ET0), precipitation (P), and δ18O and δ2H in the study area from April 1, 2023, to July 1, 2023.

(A and B) The daily variation of δ18O and δ2H in different layers.

Regional precipitation was relatively rare (only 161.1 mm) and less than the regional ET0 level during the study period (Fig. 2). To facilitate the calculation of continuous soil water evaporation losses, this study divided 10 continuous evaporation periods according to the continuity of the 16 precipitation events. The soil moisture content gradually decreased during the study period. The soil moisture content fluctuated between 1.90% and 30.98%. The soil water content often showed a significant upward trend after regional precipitation events, especially after continuous larger precipitation events.

Temporal variability on regional soil moisture δ18O

The fluctuation range of the stable hydrogen and oxygen isotopes in the soil moisture was smaller than that of regional precipitation (−15.60% to 5.71% and −104.96% to 30.81%) (Table 1). In this study, the fluctuations in hydrogen and oxygen stable isotopes of soil moisture gradually decreased with increasing soil depth. The δ18O and δ2H values of surface layer soil moisture had a relatively larger amplitude, which indicated that the surface layer soil moisture was more significantly disturbed by the external environment. The δ18O and δ2H of soil moisture in the 40–60 cm soil depth layer showed relatively depletion values, whereas, in the surface layer, they showed relatively enriched values (Table 1).

Table 1 The general characteristics of the isotopic composition of soil moisture and isotope values and precipitation corresponding to each precipitation event during the study period.

Types	Time	Depth
(cm)	Samples
number	δ18O (‰)	δ2H (‰)	Types	Precipitation	δ18O (‰)	δ2H
(‰)	
Max.	Min.	Mean	Max.	Min.	Mean	Period	Events	(mm)	
Soil moisture	2023.4.1–2023.4.30	0–20	30	−2.16	−6.65	−4.78	−38.57	−52.72	−46.33	Precipitation	P1	2023.4.3	21	−6.84	−40.98	
	20–40	30	−3.79	−7.07	−5.39	−41.48	−56.62	−51.01		2023.4.4	19	−15.60	−104.96	
	40–60	30	−3.93	−8.35	−5.77	−46.98	−58.65	−52.12	P2	2023.4.12	0.2	5.71	30.81	
	60–80	30	−3.93	−8.23	−5.53	−45.45	−59.63	−51.62	P3	2023.4.21	17.3	−5.76	−24.60	
	80–100	30	−3.35	−7.91	−5.69	−43.51	−61.04	−53.96		2023.4.23	10.72	−5.16	−23.88	
2023.5.1–2023.5.30	0–20	31	−4.24	−12.59	−6.86	−16.11	−60.48	−41.62	P4	2023.4.27	2.4	−8.90	−62.19	
	20–40	31	−3.74	−12.01	−8.10	−21.00	−61.97	−45.82	P5	2023.5.3	1.78	0.86	14.16	
	40–60	31	−4.67	−12.43	−8.85	−36.13	−59.83	−47.18	P6	2023.5.6	22.5	−4.74	−22.68	
	60–80	31	−4.11	−12.54	−8.37	−33.37	−58.83	−46.12	P7	2023.5.27	6.7	−6.92	25.35	
	80–100	31	−4.00	−11.98	−8.74	−30.04	−61.49	−46.04		2023.5.28	9.7	−5.16	−23.88	
2023.6.1–2023.6.30	0–20	30	−0.13	−8.67	−4.93	−20.47	−71.00	−38.18		2023.5.30	8.7	−4.15	−18.49	
		20–40	30	−4.22	−11.10	−7.66	−26.34	−62.86	−47.91			2023.5.31	6.5	−3.47	−15.35	
		40–60	30	−3.63	−11.03	−8.00	−42.66	−68.10	−55.43		P8	2023.6.11	0.3	−0.03	−15.69	
		60–80	30	−4.22	−11.10	−7.73	−26.34	−62.86	−48.34		P9	2023.6.18	4.3	−4.69	−35.82	
		80–100	30	−3.76	−9.62	−6.81	−48.22	−68.15	−56.02		P10	2023.6.26	5.3	−0.63	−14.77	
												2023.6.28	24.7	−9.02	−17.18	

Owing to the relatively similar temporal variability that was observed in the hydrogen and oxygen stable isotopes of regional soil moisture, only δ18O was used to discuss the temporal variation of soil moisture in this study. In the early part of the study period (2023.4.1–2023.5.6), a smaller fluctuation of the soil moisture δ18O value appeared in this period, with an average value of −5.65 ± 1.89‰, and the corresponding SWC also showed a stable trend (average value: 22.44%) (Fig. 2). From 2023.5.6 to 2023.5.14, soil moisture δ18O rapidly decreased at the beginning of this period and remained at a relatively low δ18O level, especially in the surface layer (Fig. 2). The corresponding meteorological elements also showed significant variations, and regional air temperature and ET0 showed a significant upward trend during this period (2023.5.6–2023.5.20). The relatively depleted soil moisture δ18O in this period may be related to the melting and evaporation of the seasonal permafrost in the deep soil due to the influence of surface temperature and strong ET0. The evaporated soil moisture of the deep soil layer (the depleted δ18O value) accumulated in the surface soil layer and further resulted in the dilution of soil moisture δ18O (Li et al., 2020; Song et al., 2017). Subsequently, soil moisture δ18O values gently increased (mean value: −8.54 ± 2.59‰) after week 6, while the SWC showed a slow fluctuating decline trend (average value 21.96%). From weeks 9–13 (2023.6.3–2023.7.1), soil moisture δ18O values in the study area presented notable fluctuations with a larger amplitude of the SWC variation (average value 18.95%), and the maximum soil moisture δ18O value was observed during the study period. Compared with the previous period, the average δ18O value of soil moisture in this period was more enriched (average value is −6.94 ± 2.23‰). Obviously temporal heterogeneity occurred in the soil moisture δ18O values in the study period, which was similar to the temporal variation trend of soil moisture δ18O in other regions (central and western regions) of the Chinese Loess Plateau (gradually enriched from April to June) (Wang et al., 2019; Zhang et al., 2024). In addition, the response of soil moisture δ18O variability to precipitation events during the 0–100 cm soil depth layers presented inconsistency, especially in April (Fig. 2A). After the precipitation event on April 4, the soil moisture δ18O in the 0–20 cm soil depth layer was significantly depleted the following day; the depleted soil moisture δ18O occurred 2 days later in the 20–100 cm soil depth layer. A similar phenomenon occurred after the precipitation event on April 23.

Spatial variability on regional soil moisture δ18O

Figure 3 shows the vertical spatial variation in the average values of soil moisture δ18O and δ2H values at different soil depth layers during 10 consecutive evaporation periods after precipitation events (the P1-P10). Overall, the soil moisture’s δ18O values had a “C” shaped vertical space distribution. Similar vertical spatial distribution characteristics of soil moisture’s δ18O values in the 0–100 cm depth soil layers were also observed in the central region of the Chinese Loess Plateau (Wang et al., 2019). The soil moisture δ18O values in the middle soil depth layers were relatively depleted, while relatively enriched soil moisture δ18O values appeared in the surface and bottom soil layers, which was inconsistent with the spatial variation of soil moisture’s δ18O values in the western and northern regions of the Chinese Loess Plateau (Xiang et al., 2021; Zhang et al., 2024). During the study period, the average soil moisture δ18O value was most enriched in the 0–20 cm soil depth layer, which gradually decreased with increased soil depth and reached the minimum value at the 40–60 cm soil depth layer (Fig. 3). In addition, the soil moisture δ18O was gradually enriched after the 40–60 cm soil depth layer with increased soil depth layer (Fig. 3).

Figure 3 Depth variations on δ18O and δ2H of soil moisture under different precipitation conditions.

According to the accumulated precipitation, 10 consecutive evaporation periods (the P1–P10) in this study were divided into three main types: accumulated precipitation amounts of 0–5 mm (the P2, P4, P5, P8, and P9), 20–30 mm (the P3 and P6), and 30–40 mm (the P1, P7, and P10). A similar trend of the vertical variability on soil moisture δ18O was observed during the three evaporation periods of the accumulated precipitation amount of 30–40 mm (the P1, P7, P10), which were also consistent with the vertical variation trend for the entire study period. In the five evaporation periods (the P2, P4, P5, P8, and P9) of the accumulated precipitation amount of 0.2–5 mm, the vertical spatial distribution of soil moisture δ18O values was also consistent with the entire study period except for the P9. During the P9, a relative enrichment peak value of soil moisture δ18O was observed in the 40–60 cm soil depth layer. An obviously inconsistency for the vertical spatial distribution of soil moisture δ18O occurred under 20-30mm precipitation conditions compared with the entire study period (Fig. 3), which showed relatively enrichment δ18O values at the 20–40 cm soil depth layer, whereas the relatively depletion δ18O values of the 0–20 cm soil depth layer occurred during this period. The vertical spatial heterogeneity of soil moisture δ18O was weak during the P1–P5, while the vertical spatial difference of soil moisture δ18O was gradually increased after the P6.

The relationship between δ18O and δ2H of soil moisture in different soil layers

During the study period, regional precipitation δ18O values varied from −15.60‰ to 5.71‰, with an average value of −5.03‰ and a standard deviation of ±5.11. Among them, the relatively higher standard deviation of precipitation δ18O values occurred in April (6.31‰), which indicated a more complex precipitation process in this month. The larger fluctuation of precipitation δ18O in April may have been related to regional temperature fluctuations and complex water vapor sources during this period (Craig, 1961). The meteorological water line (LMWL4-6) during the study period was fitted using δ2H = 5.25δ18O + 2.39 (R2 = 0.78, n = 16). The smaller slope of LMWL4-6 indicates the regional climate pattern was relatively arid during the study period (Sun et al., 2020c). In this study, the slopes of the soil moisture line (SWL) at different depths of soil layers were smaller than that of the LMWL4-6, which may be related to the strong evaporation of soil water (Li et al., 2023). The dispersion of the soil moisture sample points in the surface soil layer was greater and gradually decreased with increasing soil depth (Fig. 4). Compared with the other 3 months, the soil moisture samples in May were more dispersed, and significant inconsistencies appeared in the different soil layers during May. Most water moisture samples collected in April were concentrated in the middle of the LMWL4-6 and were relatively displaced in different soil layers. The locations of the soil moisture sampling points were significantly different from those in the other months (Fig. 4). Most soil moisture samples were on the upper left of the LMWM4-6 and GMWL, indicating a strong recycling process. The dispersion of the soil moisture sample points in June gradually weakened as the soil layer depth increased and almost coincided with the soil moisture samples in April in the 80–100 cm soil layer. The slope of the SWL in April generally increased with increasing soil depth. The smallest value of the SWL slope appeared at the dispersion of the soil moisture sample points in June, gradually decreased with increased soil layer depth, and almost coincided with the April soil moisture samples in the 80–100 cm soil layer, which indicated that the soil moisture samples were affected by regional evaporation during April. In May, the SWL slope was opposite to that in April, showing a gradually decreasing trend with increasing soil depth. The slope of the SWL in June was generally consistent with that in April.

Figure 4 Dual-isotope plot of precipitation and soil water at different depths.

Lc-excess of soil moisture at different soil depth layers

During the study period, the average lc-excess values in soil moisture were generally less than 0‰ owing to evaporative fractionation of stable isotopes. Mean values of soil moisture lc-excess in the soil layer at the 0–20, 20–40, 40–60, 60–80, and 80–100 cm depth soil layers were −14.88, −11.31, −14.56, −13.05, and −13.26‰, respectively. The average lc-excess value of soil moisture at the 0–20 cm soil depth layer was lower than that of the other soil layers, indicating the soil moisture at the 0–20 cm soil depth layer was more strongly disturbed by regional evaporation. The average lc-excess of soil moisture showed an increasing fluctuation trend with increasing soil depth in the study region, which was consistent with the spatial-temporal trend of lc-excess of soil moisture compared with other regions of the Chinese Loess Plateau (Xiang et al., 2021; Zhang et al., 2024) (Fig. 5).

Figure 5 Vertical distribution heat map of soil water δ2H (A), δ18O (B), and lc-excess (C) at 0–100 cm soil depth layer during the study period.

The soil moisture lc-excess showed a “low-high-low” temporal trend, which was contrary to the temporal trend of soil moisture δ18O (Fig. 5). During weeks 1–5, all the soil moisture lc-excess was below 0‰ with a slight fluctuation, indicating that the soil moisture at different depths experienced relatively stable intensity evaporation during this period. The increasing soil moisture lc-excess after week 6 (even located above 0‰) indicated the soil moisture was affected by other water sources except for rainfall during this period, which may be related to the condensation of water vapor (evaporated from surrounding vegetation soil and lakes) and being adsorbed by the topsoil at night. This season is the high-incidence period of dew formation in the Loess Plateau. The adsorption of dew (the depleted stable isotopes) by topsoil may be a possible reason for the lower soil moisture δ18O values and higher soil moisture lc-excess values during this period (Li et al., 2022a). In this period, the depth layer soil moisture also began to be affected by evaporation along with the rapid rise of surface temperature, and the accumulation of the evaporated water vapor (depleted δ18O) in the surface soil layer may also be an important reason for depleted δ18O and higher lc-excess during this period (Fig. 5). During the study period, the soil moisture lc-excess gradually decreased in the 0–20 cm soil depth layer, whereas the soil moisture lc-excess in the deeper soil layers was more stable, except in early May (Fig. 5).

Quantitative assessment on evaporation loss of soil moisture

Using the C-G model, we estimated the evaporation loss (f) of the soil layer with a 0–100 cm depth under continuous evaporation during the observation period. Significant spatial and temporal heterogeneity was observed in the soil moisture evaporation loss fraction (f) during the study period (0–23.35%) (Fig. 6). Overall, the evaporation loss of soil moisture was weak at the beginning of the study period, whereas a relatively strong soil moisture evaporation loss occurred between mid-late May and mid-June (Fig. 6). The mean values of soil moisture’s f in April, May, and June were 2.74 ± 1.06%, 11.36 ± 5.70%, and 8.41 ± 3.10%, respectively. Soil water’s evaporation loss in this study area showed inconsistent temporal variation trends compared with other regions of the Chinese Loess Plateau (Yong et al., 2020; Zhang et al., 2024), which posed challenges for the management of soil water resources in the non-monsoon period of the study area. The soil moisture evaporation loss was relatively weak at the end of precipitation, whereas a relatively larger intensity of continuous evaporation loss of soil water usually occurred at longer precipitation intervals. A relatively smaller soil moisture f was observed in the 0–40 cm soil layer, except for the P1 and P2 (Fig. 6). In the 40–80 cm soil layer, the soil moisture f at the P6 and later periods showed significant vertical spatial inconsistency (Fig. 6). At 80–100 cm soil layer, significant soil water evaporation losses occurred only in the P6 and the P8. As for temporal variation, soil moisture evaporation loss showed relatively lower and stable values during the P1 and P2 and then gradually increased. Soil moisture f increased from the P6 and reached an extremely large value on May 20.

Figure 6 Variations in soil water evaporation losses (based on δ18O) and SWC at different soil depths from 01 April 2023 to 01 July 2023.

The maximum value of soil moisture evaporation loss appeared in the 0–20 cm soil depth layer (8.97%), and the soil evaporation loss fluctuated as the soil depth increased, which was consistent with the lc-excess variation in soil moisture in “Lc-excess of Soil Moisture at Different Soil Depth Layers”. The relatively smaller soil moisture f in April (with a weak vertical spatial difference) indicated weaker regional soil moisture evaporation. A relatively high soil moisture f was observed in May, and the soil moisture evaporation intensity of the surface soil layer was significantly higher than that of the deep soil layer during this month. Significant vertical spatial heterogeneity of soil moisture evaporation loss was observed in late June, which may be related to fluctuations in the main environmental elements (precipitation, temperature, and humidity) during this month.

Determination of the relationship between soil water evaporation loss and influencing factors

Several studies have revealed that the mechanism of soil water migration in the Loess Plateau is complex, and regional soil moisture evaporation loss is evidently affected by the main environmental factors (such as air temperature, precipitation, potential evaporation, soil water content, relative humidity, and vegetation cover) (Dubbert et al., 2013; Li et al., 2020; Song et al., 2017). Because the sampling site of this study area had no evident vegetation coverage or significant groundwater recharge (the altitude of the sampling depth was higher than the regional lake surface), air temperature (T), potential evaporation (ET0), soil water content (SWC), wind speed (WS), and atmospheric relative humidity (RH) were selected as potential factors for regional soil moisture evaporation loss to conduct the correlation analysis.

The RDA results showed an inconsistency in the main controlling factors of soil moisture evaporation loss at different depths of the soil layers (Fig. 7). Overall, the f of soil moisture in the study area showed a significant correlation with T, ET0, and SWC. While a relatively weak correlation was observed between f and WS/RH. Among them, a negative correlation was observed between SWC and RH and continuous soil evaporation loss. In the 0–20 cm soil depth layers, regional ET0, WS, and T were positively correlated with soil moisture f, and the order of the correlation degree was ET0 > WS > T, indicating that the evaporation process of the surface soil layer was mainly affected by the variability in regional ET0 and WS. In the 20–60 cm soil depth layer, a positive correlation was observed between regional ET0 and regional soil moisture evaporation loss, and the influence of regional T variability on soil moisture evaporation loss showed an increasing trend. A weak correlation between ET0, WS, and T and regional soil moisture f was observed in the 60–100 cm soil depth layer, which indicated that soil layers below 60 cm depth were not disturbed by external environmental elements.

Figure 7 The RDA analysis ranking for the influencing factors on soil moisture evaporation losses at different depths of soil layers.

T, Air temperature; ET0, potential evapotranspiration; RH, relative humidity; WS, Wind speed; SWC, soil moisture content; f, soil moisture evaporation loss).

Generally, a correlation existed between the continuous evaporation loss of regional soil moisture and the above five environmental control factors; however, this correlation gradually weakened with increasing soil depth. The variation in regional air temperature had a stronger connection with the f of soil moisture, and increasing air temperature resulted in increased soil moisture f of the surface soil layer. Increased regional relative humidity reduced the evaporation strength of soil moisture by inhibiting the diffusion movement of the evaporated soil moisture, while a greater wind speed accelerated the diffusion of evaporated water vapor in the soil, increasing soil evaporation intensity, which is consistent with previous studies (Lyu & Wang, 2021). In general, SWC variability directly determines the water yield from soil moisture evaporation that can be supplied. Regional ET0, as an original driver of the regional soil moisture evaporation process, also significantly impacts the strength of the regional soil moisture evaporation loss.

To further reveal the internal relationship between the continuous evaporative loss of soil moisture and environmental elements, a multiple linear regression analysis model was used to construct a driving mechanism model for continuous soil moisture evaporation under different regional precipitation conditions. According to the results in the previous sections, the soil moisture evaporation loss fraction was defined as the independent variable (y) in this study, and five environmental elements were defined as dependent variables (x1, x2, x3, x4, and x5), which were analyzed using MLR regression under different precipitation conditions (the 0.2–5, 20–30, and 30–40 mm, respectively). Table 2 shows the results of the evaporation loss driving model of soil moisture at different soil layer depths under different precipitation conditions, with relatively large R2 fluctuations varying from 0.14 to 0.95.

Table 2 The relationship between soil moisture evaporation loss and influencing factors under different cumulative precipitation levels in the study area from April 1, 2023, to July 1, 2023.

Accumulated precipitation (mm)	Soil depth (cm)	Evaporation model	R2	
0.2–5	0–20	y = −6.46 + 0.07x1 + 1.28x2 + 0.03x3 − 0.51x4 + 0.09x5	0.30	
20–40	y = 0.17 + 0.27x1 − 0.25x2 − 0.09x3 + 0.21x4 + 0.05x5	0.14	
40–60	y = −3.8 + 0.27x1 + 0.02x2 + 0.25x3 − 0.057x4 + 0.03x5	0.22	
60–80	y = −12.52 + 0.28x1 + 0.68x2 + 0.27x3 − 0.56x4 + 0.08x5	0.24	
80–100	y = 17.02 − 0.32x1 + 1.52x2 − 0.57x3 − 1.79x4 + 0.11x5	0.27	
0–100	y = −1.33 + 0.08x1 + 0.61x2 − 0.03x3 − 0.48x4 + 0.07x5	0.37	
20–30	0–20	y = −53.01 + 0.64x1 + 11.55x2 − 0.62x3 − 7.50x4 + 0.41x5***	0.76	
20–40	y = −39.47 + 0.57x1 + 7.80x2 − 0.24x3 − 4.41x4 + 0.28x5***	0.73	
40–60	y = −35.65 + 0.50x1 + 8.66x2 − 0.67x3 − 5.21x4 + 0.31x5*	0.58	
60–80	y = −20.62 + 0.36x1 + 5.87x2 − 0.65x3 − 1.54x4 + 0.23x5	0.44	
80–100	y = −6.17 + 0.38x1 + 3.11x2 − 0.60x3 + 0.07x4 + 0.13x5	0.3	
0–100	y = −29.53 + 0.53x1 + 7.17x2 − 0.65x3 − 3.61x4 + 0.27x5**	0.66	
30–40	0–20	y = 19.43 + 0.07x1 − 0.76x2 − 0.77x3 + 1.77x4 − 0.03x5*	0.74	
20–40	y = 10.25 + 0.03x1 + 2.04x2 − 0.57x3 − 2.65x4 + 0.05x5***	0.95	
40–60	y = 9.47 + 0.02x1 + 0.13x2 − 0.45x3 + 1.47x4 − 0.04x5	0.55	
60–80	y = 7.34 − 0.04x1 + 3.05x2 − 0.40x3 − 3.81x4 + 0.08x5*	0.88	
80–100	y = −1.74 + 0.13x1 − 0.10x2 + 0.14x3 − 0.72x4 + 0.04x5	0.27	
0–100	y = 13.02 − 0.002x1 + 0.73x2 − 0.53x3 − 0.56x4 + 0.001x5*	0.88	
Notes:

* Indicates that the correlation passes the significance test at 0.05.

** Indicates that the correlation between the two is at the 0.01 level and passed the significance test.

*** Indicates that the correlation between the two is at the 0.001 level and passed the significance test.

x1 is T, x2 is ET0, x3 is SWC, x4 is WS, and x5 is RH.

At an accumulated precipitation amount of 0.2–5 mm, the R2 of the fitted driving models of all soil depth layers did not exceed 0.4 and P > 0.05, which indicated that the fitting effect of the soil moisture evaporation driving model was not significant. A statistically significant environment-driven model for continuous evaporation of the 0–100 cm soil depth layer was fitted under an accumulated precipitation amount of 20–30 mm. Among them, a significant linear relationship between the continuous evaporation of soil moisture and the mainly environmental elements appeared in soil layers of 0–20 and 20–40 cm depths, respectively.

At an accumulated precipitation amount of 20–30 mm, the fitting equation of the soil moisture evaporation driving model for the 0–100 cm soil depth layer was significant (R2 = 0.88, P < 0.05). The soil moisture evaporation driving model for the 20–40 cm soil depth layer was particularly significant (R2 = 0.95, P < 0.001). With increasing soil depth, the linear relationship between the continuous evaporation of soil moisture and environmental elements gradually weakened. In this study, depth soil evaporation showed a relatively weak response to variation of environmental elements (Fig. 7 and Table 2), which may be related to the following reasons. First, depth layer soil moisture fluctuated less than other soil layers, and the exchange of matter and energy with the surface usually was relatively weak (Du et al., 2021). Second, the influence of air temperature has obvious hysteresis on the deep soil layer, and its influence to the depth layer is weak than that of other soil layer (Lyu & Wang, 2021). Third, precipitation affected the soil moisture f through infiltration, while the depth soil received less surface precipitation than surface soil layers (Piayda et al., 2017). In summary, the environmentally driven model of the continuous evaporation of soil moisture was not statistically significant for relatively small rainfall amounts but was suitable for larger amounts, especially for the surface soil layers. However, this study has some limitations due to the observed data of this study just only collected from a single point during 1 year.

Conclusions

In this study, we presented daily soil moisture stable isotope in different soil layer depths during the growing period of the non-monsoon season, and quantitatively estimated the rate of continuous evaporation loss of regional soil moisture under different precipitation conditions and investigated its controlling factors in the ECLP.

In this study, we presented the daily observations of δ2H, δ18Op, of soil moisture in different soil layer depths during the growing period of the non-monsoon season in the ECLP. We quantitatively estimated the rate of continuous evaporation loss of regional soil moisture under different precipitation conditions and investigated the factors controlling the evaporation of regional soil moisture in the ECLP.

During the study period, the main meteorological elements and soil moisture showed significant temporal fluctuations, which resulted in a significant temporal heterogeneity in the soil moisture δ18O values. The response of soil moisture δ18O variability to precipitation events during the 0–100 cm soil depth layers showed inconsistency, especially in April.

The δ18O and δ2H values of surface layer soil moisture had a relatively larger amplitude and gradually depleted with the increase of soil depth. A “C” shaped vertical space distribution was observed in the soil moisture δ18O values. The “low-high-low” temporal trend of soil moisture lc-excess was contrary to the temporal trend of soil moisture δ18O.

A significant spatial and temporal heterogeneity was observed in the soil moisture evaporation loss fraction (f) during the study period (0–23.35%), with weaker values at the beginning of the study period and larger values between mid-late May and mid-June. The soil moisture evaporation loss was relatively weak at the end of precipitation, whereas a relatively larger intensity of continuous evaporation loss of soil water usually occurred at longer precipitation intervals.

The continuous evaporation loss of soil moisture in the study area showed a significant correlation with T, ET0, and SWC and a relatively weak correlation with WS and RH. The environmentally driven model of the continuous evaporation of soil moisture is suitable for larger amounts, especially for the surface soil layers.

Supplemental Information

Supplemental Information 1 Raw data.

Additional Information and Declarations

Competing Interests

Author Contributions

Data Availability

The authors declare that they have no competing interests.

Congjian Sun conceived and designed the experiments, analyzed the data, prepared figures and/or tables, authored or reviewed drafts of the article, and approved the final draft.

Sitong Meng performed the experiments, analyzed the data, prepared figures and/or tables, and approved the final draft.

Wei Chen analyzed the data, authored or reviewed drafts of the article, and approved the final draft.

The following information was supplied regarding data availability:

The raw measurements are available in the Supplemental File.

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
