# Peer review of "Fluctuations of continuous soil moisture evaporation under different rainfall conditions during the growing period of the non-monsoon season, the eastern Loess Plateau"

_PeerJ, doi:10.7717/peerj.18514_

## Round 0.1 · original submission · Major Revisions

You manuscript has been reviewed by two experts in this research field: one of them recommends a major revision and another suggests a rejection. These comments might usually lead to a simple rejection. However, I would like give you an opportunity to modify it if you think you can address the review comments. I will ask the same reviewers to have another round of review.

Reviewer 1 ·

Basic reporting

The manuscript systematically studied the fluctuation of soil moisture evaporation under different rainfall conditions during the non-monsoon growing season, which is of great significance for understanding regional water cycle and agricultural water resource management.

The manuscript is written in clear and professional English. The language is mostly unambiguous, allowing for easy understanding of the content.

The introduction provides a solid background, showing the context and importance of soil moisture in the Eastern Loess Plateau (ECLP). The literature is well referenced and relevant, highlighting the significance of soil moisture evaporation in agricultural sustainability and regional water resource management.

The structure need be improved. The current manuscript presents many results, but there is no clear discussion, which makes the article less like an academic article

The figures included in the manuscript are relevant. However, the Figures need further modification as the fonts are too small and appear unclear. Some figures lack labels a, b, c. More importantly, the absence of legends, such as the meaning of different colored shadows in Figure 2. Further clarification in the figure could enhance the reader's understanding.

Experimental design

The use of the Craig-Gordon model for quantifying evaporation loss through long-term observations of stable isotopes in soil moisture is scientifically sound, offering a robust dataset. However, the manuscript would benefit from more explicit details on the sampling frequency (daily or every two days), procedure for soil water extraction and statistical analysis methods to ensure full transparency and reproducibility.

Validity of the findings

The manuscript used stable isotopes of hydrogen and oxygen to calculate soil evaporation, and this method is effective and has been reported in some studies. The research results indicate a significant correlation between soil water evaporation loss and environmental factors, which provides important information for regional water resource management.

All underlying data have been provided; they are robust, statistically sound, and controlled.

Additional comments

1. Please provide the source of meteorological data.

2. the calculation of the evaporation loss ratio in the manuscript did not specify which isotope (δ2H or δ18O) was used.

3. L219, the current Vienna Standard Mean Ocean Water sample code is V-SMOW2, not V-SMOW.

4. L236, the definition of lc-excess lacks references.

5. L424~425, the local atmospheric water line (LMWL) presented in the manuscript only regresses through precipitation samples from April to June and cannot be referred to as the LMWL. It cannot represent the complete characters of local climate.

6. L433~434, the higher the evaporation intensity, the lower the remaining soil moisture content may be incorrect, which is also related to the amount of water entering the soil

7. L478~479, isotopic values description is usually determined by “enriched or depleted”, not “low or high”.

8. L574 and L578~579, the equation has been presented in the table and does not need to be repeated

9. Suggest further discussion on why evaporation in deeper soil layers is less associated with environmental factors.

Reviewer 2 ·

Basic reporting

English needs improvement.
Lack of clear and novel hypotheses.

Experimental design

The study fails to present novel methods or ideas, it merely applies existing techniques to a case study.

Validity of the findings

All information is presented, but can be improved. No questions or hypotheses are answered because they were never described in the first place.

Additional comments

The manuscript "Fluctuations of continuous soil moisture evaporation under different rainfall conditions during the growing period of the non-monsoon season, the Eastern Loess Plateau" by Sun et al. presents a detailed common analysis of the spatiotemporal soil isotopic composition of bare soil (0-100 cm) during the non-monsoon season, including an analysis of soil evaporation losses using the Craig-Gordon model. Unfortunately, the paper does not introduce any novel concepts, ideas, or tools/methods, but rather offers a contribution by applying a standard technique (the Craig-Gordon model) to a specific case study on the eastern Loess Plateau. Thus, the results presented in the manuscript do not provide particularly groundbreaking insights into the field of isotope hydrology. The study largely reiterates or confirms findings from similar investigations, lacking a significant degree of originality. Additionally, the manuscript could benefit from further refinement of the English language in the text.
Some comments:
The article's length could be reduced by eliminating unnecessary text, particularly in the introduction and results/analysis sections.
Line 222: "In this study, soil samples were collected continuously in the study area." Please clarify if this means daily collection.
Line 431: "P1-P10. "divided into three main types: accumulated precipitation amounts of 0 mm (P2, P4, P5, P8, and P9), 20–30 mm (P3 and P6), and 30–40 mm (P1, P7, and P10)." Consider presenting the results by grouping and averaging the data. In addition, there is an inconsistency since the periods are shown in Figure 6, but this analysis is presented in Figure 3.
Figure 2: Please clarify the meaning of the different background colors.
Figure 3: This analysis could be simplified to show only the mean ± standard deviation of the three established groups.
Figure 4: The graph's presentation should be revised. Consider creating three dual plots corresponding to each month. In each dual plot, display all data points for soil data at all depths, stratified by color palette. The "The atmospheric precipitation line in the research area" plot can be omitted as it is already presented in each dual plot. Additionally, please add labels (a), (b), and (c) to each plot.

---

## Round 0.2 · Minor Revisions

Thank you for the revision of the manuscript and your patience in the review process so-far. I have been asked to take over as the prior Academic Editor was unavailable.

The quality of the manuscript has been improved. However, there is still an open issue with chapter 3, which lacks a clear discussion (see comment from reviewer 1). I propose to change the heading of the chapter to "Results and discussion". In addition, you must try to critically examine and classify your own findings against the background of the scientific literature, which should be used to a greater extent for comparisons

Reviewer 1 ·

Basic reporting

After carefully reviewing the revised manuscript, I believe that the author's revisions have tracked and addressed most of the issues raised in the previous round of review.

Experimental design

no comment. After revision, the methods now have sufficient details and information.

Validity of the findings

This article does not make any new discoveries or propose any new methods. It is a specific case study of a mature method (Craig Gordon model) in the eastern Loess Plateau.This method is effective and has been reported in many studies.

For this manuscript, all underlying data have been provided; they are robust, statistically sound, and controlled.

Additional comments

The current manuscript still lacks clear discussions, which need be improved. The analysis lacks depth and logical coherence, is poorly organized, and most of the space is focused on descriptive results, with almost no literature support.

---

## Round 0.3 · accepted · Accept

Thank you for the revision of the manuscript. Although the revision of the discussion could have been more extensive, I hereby certify that you have adequately taken into account our comments and improved the manuscript accordingly. Based on my assessment as an Academic Editor, your manuscript is now ready for publication.